# Changes in Serum Immunoglobulin G Subclasses during the Treatment of Patients with Chronic Obstructive Pulmonary Disease with Infectious Exacerbations

Thang Ba Ta [1,†], Tien Tran Viet [2], Kien Xuan Nguyen [3,†], Cong Hai Nguyen [4], Hoan Ngoc Vu [1], Tuan Dinh Le [5], Son Tien Nguyen [5], Hung Khac Dong [1], Nhung Kim Thi Pham [1] and Bang Ngoc Dao [1,*]

1 Respiratory Center, Military Hospital 103, Medical Military University, Hanoi 10000, Vietnam
2 Department of Infectious Diseases, Military Hospital 103, Medical Military University, Hanoi 10000, Vietnam
3 Department of Military Medical Command and Organization, Medical Military University, Hanoi 10000, Vietnam
4 Department of Pneumology, Military Hospital 175, Ho Chi Minh City 71409, Vietnam
5 Department of Rheumatology and Endocrinology, Military Hospital 103, Medical Military University, Hanoi 10000, Vietnam
* Correspondence: daongocbang@vmmu.edu.vn
† These authors contributed equally to this work and share first authorship.

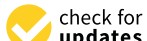



**Highlights:**

**What are the main findings?**

- The levels of IgG1, IgG2, IgG3, and IgG4 are reduced in patients with COPD with acute infectious exacerbations.
- IgG3 levels are related to the severity of COPD.

**What are the implications of the main findings?**

- IgGs are important in follow-up studies to assess COPD exacerbations.
- This study could be a suggestion for conducting future studies using immunotherapy to treat COPD.

**Abstract:** Introduction: Despite the theoretical importance of serum immunoglobulin (Ig) in the outcome of COPD exacerbations, the existing evidence for this has not been enough. This study was performed to evaluate changes in serum Ig levels and their relationship with outcomes of acute infectious exacerbations in patients with COPD. Methods: The prospective study was conducted at Military Hospital 103 from August 2017 to April 2019. Group D patients with COPD with infectious exacerbation were selected for participation in the study. The control group consisted of 30 healthy people. The patients were provided clinical examination and laboratory service; simultaneously, we measured their serum Ig levels (total IgG, IgG1, IgG2, IgG3, IgG4) at two time points: at admission (T1) and the final health outcome (T2). Results: The median levels of total IgG in patients at times T1 and T2 were significantly lower compared with those in the healthy group (1119.3 mg/dL and 1150.6 mg/dL compared with 2032.2 mg/dL) ($p < 0.001$). Regarding changes among IgG subclasses, the IgG1, IgG3, and IgG4 levels measured at T1 and T2 were reduced significantly compared with the control group ($p < 0.05$); the IgG3 levels at T1 were significantly higher than those at T2. IgG3 levels in patients with life-threatening exacerbations were significantly lower than the remaining ones (24.6 (26.8–155.5) mg/dL and 25.6 (29.5–161.2) mg/dL, respectively, $p = 0.023$). Conclusions: In group D patients with COPD with infectious exacerbations, there was a decrease in the serum IgG, IgG1, IgG3, and IgG4 levels. IgG3 levels were associated with the severity of COPD exacerbation.

**Keywords:** immunoglobulin G subclasses; chronic obstructive pulmonary disease (COPD); infectious exacerbation

## 1. Introduction

Chronic obstructive pulmonary disease (COPD) is currently a global burden [1]. According to global and regional estimates of COPD prevalence (2015), there were more than 230 million COPD cases among urban dwellers (prevalence of 13.6%) and 153.7 million COPD cases among rural dwellers (prevalence of 9.7%), the prevalence in men aged 30 years or more is 14.3% compared to 7.6% in women [2]. Currently, among the most common specific causes of death, the global mortality of COPD cases is the fourth, and this is predicted to climb to the third by 2030 [3]. In Vietnam, according to the estimation of the Asia Pacific Respiratory Society in 2003, the prevalence of COPD was 6.7% and was the highest in 12 Asian countries [4].

In COPD, a severe event is an exacerbation, which worsens the disease and quality of life and increases mortality in patients [5–7]. A common cause of an exacerbation is a bacterial infection of the lower respiratory tract [8,9]. During an infectious exacerbation, local and systemic inflammatory responses are amplified [10]. The level of inflammation depends on the body's immune system and the types of infectious agents, the status of a patient's immune system also affects the patient's infection in exacerbation [11,12]. Recently, several studies have shown that immunoglobulin (Ig) plays an important role in respiratory tract infections in patients with COPD [13,14]. A decrease in serum Ig levels in stable patients with COPD can potentially raise the frequency and severity of exacerbations and recurrent respiratory infections and increase their risk of mortality [15–17]. Ig replacement therapy in patients with COPD has been studied and has shown good initial results [18]. In Vietnam, the number of COPD patients is increasing, and the rate of re-hospitalization and death due to infections is relatively high, burdening the healthcare system. Thus, the investigation of serum Ig levels might positively contribute to the treatment and management of patients with COPD. However, there is currently very little research on the role of Ig in COPD. Therefore, this study was conducted to evaluate changes in serum Ig levels and their relationship with the outcomes of patients with COPD with acute infectious exacerbations.

## 2. Methods

The study was conducted on 97 group D patients with COPD who had infectious exacerbations and were under treatment at Vietnam Military Hospital 103, Hanoi, Vietnam, from August 2017 to April 2019. Criteria of patient selection were as follows: patients with COPD with post-bronchodilator $FEV_1/FVC < 70\%$ who were outpatients of the Respiratory Center of Vietnam Military Hospital 103, Hanoi, Vietnam, before the exacerbation. Patients are classified as group D according to the Global Initiative for COPD guidelines [19]. An exacerbation was defined if a patient had two of the following symptoms: increased dyspnea, increased sputum production, and purulent sputum [20]. Evaluation of an infectious exacerbation was based on the following criteria: fever, purulent sputum production, and positive sputum culture. All patients had not received systemic corticosteroids within 30 days prior to the day of hospital admission. Exclusion criteria included patients with combined diseases such as asthma, autoimmune diseases, malignancy, congestive heart failure, pulmonary tuberculosis, and acute viral infections and who received systemic corticosteroid treatment within 30 days beforehand.

The sample size (*n*) of 97 patients with COPD was calculated according to the "estimate a population proportion" formula used for qualitative analysis in descriptive and analytical research as follows: $n = [Z^2_{(1-\alpha/2)} \times p(1 - p)]/\Delta^2$, where $p = 0.549$ is the proportion of patients re-hospitalized due to COPD exacerbation in 12 months of follow-up [20], $Z^2_{(1-\alpha/2)} = 1.96$ is the value corresponding to $\alpha = 0.05$, and $\Delta$ is the desired deviation between the rate obtained from our study sample and *p* of 0.549 of the previous study population. In our study, $\Delta = 0.1$. The control group was composed of 30 healthy individuals aged $\geq 45$ years without any specific chronic diseases or infections. They were selected for measuring serum Ig levels as reference values for the test result in patients. The study was

approved by the Ethics Committee of Military Hospital 103 (No. 57/2014/VMMU-IRB). Consent and commitment forms were signed by the patients in the study.

The patients received clinical examinations and laboratory services such as complete blood count, C-reactive protein (CRP), procalcitonin (PCT), chest X-ray, arterial blood gas, and serum Ig concentration at 2 timepoints: the first one was when the patients were admitted to the hospital (T1) and the second one was at the time of main outcomes achieved including recovery or near the end of life (T2). Particularly, sputum bacterial culture was performed at time T1. The patients' exacerbations were treated according to the guidelines of the Vietnam Ministry of Health (2017).

Levels of serum total IgG and IgG subclasses (i.e., IgG1, IgG2, IgG3, IgG4) were measured by immunofluorescence technique with the Invitrogen kit of Thermo Fisher Scientific (Viennna, Austria) at the immunology laboratory at the Vietnam Military Medical University, Hanoi, Vietnam. The mean value of Ig levels in the control group was used as a reference value to evaluate the change in Ig concentration in patients with COPD.

The severity of the exacerbation was assessed according to the GOLD criteria (2017) [19] as follows: group I: no respiratory failure; group II: acute respiratory failure, non-life-threatening; and group III: acute respiratory failure, life-threatening. Patients with COPD were treated according to GOLD 2017 guidelines [19]. Mild exacerbations were treated with short-acting bronchodilators (SABDs). Moderate exacerbations were treated with short-acting bronchodilators (SABDs) in combination with antibiotics and/or oral corticosteroids (ICS), and oxygen therapy. Severe and critical exacerbations were treated with bronchodilators, systemic corticosteroids, antibiotics, oxygen therapy, and ventilatory support. After the exacerbation was stabilized, the patient was discharged from the hospital with maintenance therapy starting with LABA/LAMA; if there was an early exacerbation, ICS was added. In the study, all participants who were hospitalized with acute exacerbations were given systemic corticosteroids at a dose of 1 mg/kg/24 h from the beginning, with dose adjustment according to the individual's response to the treatment during their hospitalization.

The remission of the exacerbation was evaluated according to GOLD criteria: the patient can walk in the ward (if the patient was able to walk before admission); dyspnea does not affect sleeping quality; there is no need to use short-acting inhaled β2 agonists every 4 h; clinical symptoms and arterial blood gases are stable over 12–24 h. Outcomes of treatment for exacerbations were divided into two groups: good (the patient was out of exacerbation and discharged from the hospital) and dead (the patient died from COPD exacerbation) outcomes.

*Statistical Analysis*

All data were expressed as mean ± standard deviation or median (interquartile range, Q1–Q3) or percentage where available. Differences between groups were examined with either Student's *t*-test or the Mann–Whitney test. Correlations between Ig and other variables were evaluated with Spearman's correlation analysis. A *p*-value < 0.05 was defined as statistically significant. Data were analyzed with SPSS version 26 (SPSS Inc., Chicago, IL, USA).

## 3. Results

The mean age of the patients was 72.3 ± 8.1 years. Male patients accounted for 96.9%. The mean duration of COPD was 6.8 ± 4.7 years. The percentage of smoking patients was 88.6%, with an average of 18.4 ± 9.2 pack-years. Regarding the severity of exacerbation, no respiratory failure accounted for the highest rate (47.4%), followed by life-threatening acute respiratory failure and non-life-threatening acute respiratory failure (43.3% and 9.3%, respectively). Respiratory failure and chronic heart failure made up 52.6% and 24.7% in the same order. The percentage of positive sputum culture in patients was 27.8%, in which 14 patients had Gram-positive and 13 had Gram-negative infections; *S. pneumoniae* infections occurred at the highest rate (10.3%). The median serum CRP and PCT levels were

22.6 mg/dL and 0.112 ng/mL, respectively. Regarding exacerbation outcomes in patients with COPD, 69.1% of patients recovered and 30.9% died. The $FEV_1$, FVC, $FEV_1$/FVC, $PaO_2$, $SaO_2$, and pH indices of the control group were significantly higher than those of COPD patients. The indices of white blood cell count, $PaCO_2$, and CRP were higher in patients with COPD than in the control group (Table 1).

**Table 1.** The demographics, biomedical characteristics, and treatment outcomes of the study population.

| Indices | Patients (*n* = 97) | Control Group (*n* = 30) | *p* |
|---|---|---|---|
| Age: $\overline{X} \pm SD$, (min–max), years | 72.3 ± 8.1 (52–87) | 64.2 ± 7.4 (51–76) | >0.05 |
| Gender: *n* (%): | | | |
| -    Male | 94 (96.9%) | 28 (93.3%) | >0.05 |
| -    Female | 3 (3.1%) | 2 (6.7%) | >0.05 |
| $FEV_1$: $\overline{X} \pm SD$, %pred | 44.5 ± 21.4 | 94.2 ± 7.8 | <0.01 |
| FVC: $\overline{X} \pm SD$, %pred | 76.2 ± 26.8 | 95.8 ± 5.7 | <0.01 |
| $FEV_1$/FVC: $\overline{X} \pm SD$, % | 44.9 ± 11.1 | 77.9 ± 6.5 | <0.01 |
| Obstructive severity (GOLD 2017) *n* (%): | | | |
| -    GOLD 1: | 6 (6.2%) | | |
| -    GOLD 2: | 30 (30.9%) | NA | NA |
| -    GOLD 3: | 33 (34%) | | |
| -    GOLD 4: | 28 (28.9%) | | |
| Smoking status: | | | |
| -    Smoked: *n* (%) | 86 (88.6%) | Non-smoker | NA |
| -    Pack-year index: | 18.4 ± 9.2 | | |
| $\overline{X} \pm SD$, (min–max) | (10.5–30.0) | | |
| Duration of the COPD disease: $\overline{X} \pm SD$, (min–max), years | 6.8 ± 4.7 (1–20) | NA | NA |
| Severity of exacerbation (GOLD 2017): *n* (%) | | | |
| -    No respiratory failure | 46 (47.4%) | | |
| -    Acute respiratory failure, non-life-threatening | 9 (9.3%) | NA | NA |
| -    Acute respiratory failure, life-threatening | 42 (43.3%) | | |
| Complete blood count: | | | |
| -    White blood cells, $\overline{X} \pm SD$, G/L | 12.8 ± 6.9 | 7.3 ± 2.4 | <0.01 |
| -    Red blood cells, $\overline{X} \pm SD$, T/L | 4.5 ± 0.66 | 4.2 ± 0.4 | >0.05 |
| -    Platelets, $\overline{X} \pm SD$, G/L | 249.1 ± 96.3 | 223.5 ± 36.2 | >0.05 |
| Arterial blood gas test: | | | |
| -    $PaO_2$, $\overline{X} \pm SD$, mmHg | 94 ± 47.7 | 98.7 ± 1.2 | <0.05 |
| -    $PaCO_2$, $\overline{X} \pm SD$, mmHg | 50.5 ± 19.1 | 37.8 ± 2.5 | <0.01 |
| -    $SaO_2$, $\overline{X} \pm SD$, % | 93.3 ± 8.1 | 97.4 ± 2.3 | <0.05 |
| -    pH, $\overline{X} \pm SD$ | 7.36 ± 0.09 | 7.42 ± 0.03 | >0.05 |
| Complications: *n* (%) | | | |
| -    Respiratory failure | 51 (52.6%) | NA | NA |
| -    Chronic heart failure | 24 (24.7%) | | |

**Table 1.** *Cont.*

| Indices | Patients (*n* = 97) | Control Group (*n* = 30) | *p* |
|---|---|---|---|
| Result of sputum bacterial culture: *n* (%) | | | |
| -     Positive | 27 (27.8%) | | |
|     + *S. pneumoniae* | 10 (10.3%) | | |
|     + *S. aureus* | 4 (4.1%) | NA | NA |
|     + *H. influenzae* | 4 (4.1%) | | |
|     + *P. aeruginosa* | 6 (6.2) | | |
|     + *K. pneumoniae* | 3 (3.1) | | |
| -     Negative | 70 (72.2%) | | |
| Serum inflammatory markers: Median (Q1–Q3) | | | |
| -     CRP (mg/dL) | 22.6 (3.5–62.24) | 3.4 (2.1–5.7) | <0.01 |
| -     PCT (ng/mL) | (0.05–0.66) | NA | NA |
| Outcome of treatment: | | | |
| -     Good | 67 (69.1%) | NA | |
| -     Death | 30 (30.9%) | | |

IgG3 levels were negatively associated with increasing PCT ($r = -0.254$, $p = 0.012$). There was no significant correlation between IgG and CRP levels (Table 2).

**Table 2.** Correlation between serum IgG concentrations and procalcitonin or C-reactive protein at time T1.

| Ig (mg/dL) | CRP (mg/dL) | | PCT (ng/mL) | |
|---|---|---|---|---|
| | r | *p* | r | *p* |
| IgG | 0.13 | 0.20 | −0.08 | 0.46 |
| IgG1 | 0.018 | 0.865 | −0.05 | 0.60 |
| IgG2 | 0.193 | 0.059 | 0.119 | 0.244 |
| IgG3 | −0.127 | 0.214 | −0.254 | 0.012 |
| IgG4 | 0.198 | 0.051 | −0.119 | 0.242 |

The levels of total IgG in patients at times T1 and T2 were significantly lower than those in the control group (1119.3 mg/dL and 1150.6 mg/dL compared with 2032.2 mg/dL, $p < 0.001$). Regarding IgG subclasses, IgG1, IgG3, and IgG4 levels at times T1 and T2 were reduced significantly compared with those in the control group ($p < 0.05$); the IgG3 levels at time T1 were significantly higher than those at time T2 (Table 3).

The levels of IgG2 were significantly higher in the group of patients with fever than in the counterpart ($p < 0.05$). IgG1 levels were significantly higher in the group with respiratory failure than in the group without respiratory failure ($p = 0.01$). IgG3 levels in patients with life-threatening exacerbation were significantly lower than those in patients without it. In addition, the median level of IgG subclasses when the patients were admitted to the hospital did not differ according to the treatment outcomes, duration of COPD, number of COPD exacerbations per year, white blood cells, obstructive severity, or sputum bacterial culture ($p > 0.05$) (Table 4).

**Table 3.** Changes in IgG levels during the treatment of COPD exacerbations and comparison with a matched control group.

| Concentration (mg/dL) | T1 [1] (n = 97) | T2 [2] (n = 67) | Control [3] (n = 30) | p |
|---|---|---|---|---|
| IgG Median (Q1–Q3) | 1119.3 (350.5–6242.2) | 1150.6 (269.6–4519.8) | 2032.2 (1062.5–5325.8) | p [1;2] > 0.05<br>p [1;3] < 0.001<br>p [2;3] < 0.001 |
| IgG1 Median (Q1–Q3) | 367.8 (474.1–679.8) | 351.6 (507.3–680.3) | 889.8 (1293.6–1749.8) | p [1;2] = 0.239<br>p [1;3] < 0.001<br>p [2;3] < 0.001 |
| IgG2 Median (Q1–Q3) | 313.4 (490.4–715.4) | 296.7 (469.5–743.0) | 401.9 (474.0–547.0) | p [1;2] = 0.931<br>p [1;3] = 0.304<br>p [2;3] = 0.725 |
| IgG3 Median (Q1–Q3) | 25.5 (28.0–155.3) | 23.5 (55.9–174.6) | 112.0 (131.6–154.9) | p [1;2] = 0.012<br>p [1;3] = 0.002<br>p [2;3] = 0.03 |
| IgG4 Median (Q1–Q3) | 31.2 (52.9–96.1) | 22.1 (46.3–103.8) | 68.5 (92.0–115.0) | p [1;2] = 0.934<br>p [1;3] = 0.01<br>p [2;3] = 0.004 |

The subscript numbers denote the number of the group for comaprison only.

**Table 4.** Changes in IgG levels at time T1 according to the severity of clinical characteristics and outcome of treatment.

| Classification | | IgG1 (mg/dL) Median (Q1–Q3) | IgG2 (mg/dL) Median (Q1–Q3) | IgG3 (mg/dL) Median (Q1–Q3) | IgG4 (mg/dL) Median (Q1–Q3) |
|---|---|---|---|---|---|
| Fever | No (n = 71) | 362.0 (471.7–671.5) | 271.9 (462.7–690.1) | 25.6 (29.8–161.2) | 30.7 (52.3–95.9) |
| | Yes (n = 26) | 368.4 (489.5–803.0) | 456.6 (588.1–838.9) | 24.7 (26.4–139.2) | 31.6 (60.5–118.8) |
| | p * | 0.742 | 0.046 | 0.255 | 0.415 |
| Duration of the COPD | <5 years (n = 39) | 381.6 (476.8–671.5) | 338.5 (526.8–690.1) | 25.6 (36.1–155.3) | 35.2 (58.3–102.1) |
| | ≥5 years (n = 58) | 355.9 (474.1- 745.0) | 271.9 (467.8–788.8) | 25.1 (27.4–158.9) | 24.2 (51.4–95.9) |
| | p * | 0.707 | 0.649 | 0.535 | 0.681 |
| Number of COPD exacerbations | <2 times per year (n = 6) | 391.1 (417.5–469.7) | 146.5 (426.2–472.3) | 25.1 (38.9–88.5) | 25.0 (38.4–70.0) |
| | ≥2 times per year (n = 91) | 362.0 (483.2–690.9) | 313.4 (540.6–717.8) | 25.5 (28.0–158.9) | 31.2 (57.6–96.9) |
| | p * | 0.686 | 0.072 | 0.515 | 0.222 |
| Severity of exacerbation | Non-life-threatening (n = 55) | 379.2 (471.7–703.4) | 282.2 (456.0–634.0) | 25.6 (29.5–161.2) | 28.5 (47.1–77.8) |
| | Life-threatening (n = 42) | 360.5 (493.9–671.5) | 370.9 (597.9–788.8) | 24.6 (26.8–155.3) | 33.7 (69.0–135.6) |
| | p * | 0.99 | 0.508 | 0.023 | 0.347 |

**Table 4.** *Cont.*

| Classification | | IgG1 (mg/dL) Median (Q1–Q3) | IgG2 (mg/dL) Median (Q1–Q3) | IgG3 (mg/dL) Median (Q1–Q3) | IgG4 (mg/dL) Median (Q1–Q3) |
|---|---|---|---|---|---|
| Respiratory failure | Yes (*n* = 51) | 350.6 (466.0–653.8) | 282.2 (456.3–573.0) | 25.1 (27.6–152.9) | 29.1 (48.0–75.9) |
| | No (*n* = 46) | 386.4 (595.9–1014.2) | 327.3 (488.3–706.7) | 25.6 (101.7–160.5) | 32.9 (64.4–108.6) |
| | *p* * | 0.010 | 0.273 | 0.351 | 0.111 |
| White blood cells | Normal (*n* = 44) | 382.3 (472.9–743.6) | 313.5 (475.7–625.7) | 25.7 (36.1–188.7) | 29.0 (52.6–93.7) |
| | Increase (*n* = 53) | 350.0 (482.9–668.1) | 313.4 (549.8–768.5) | 25.1 (27.5–139.2) | 33.7 (57.6–96.9) |
| | *p* * | 0.417 | 0.535 | 0.223 | 0.589 |
| Sputum bacterial culture | Negative (*n* = 70) | 379.2 (478.7–703.4) | 313.4 (490.1–717.6) | 25.6 (30.7–158.9) | 33.7 (58.6–96.9) |
| | Positive (*n* = 27) | 269.3 (469.7–662.1) | 282.2 (524.5–700.9) | 24.3 (25.9–139.2) | 17.2 (44.3–96.1) |
| | *p* * | 0.759 | 0.169 | 0.483 | 0.068 |
| Obstructive severity | GOLD 1 (*n* = 6) | 418.2 (470.2–745.0) | 190.2 (484.5–573.0) | 25.1 (27.8–152.9) | 35.2 (59.5–77.8) |
| | GOLD 2 (*n* = 30) | 339.9 (442.8–690.9) | 313.4 (544.1–715.4) | 25.5 (40.6–158.9) | 27.4 (64.1–115.1) |
| | GOLD 3 (*n* = 33) | 367.8 (474.1–653.8) | 282.2 (449.3–605.7) | 24.3 (25.6–154.8) | 31.0 (42.2–67.0) |
| | GOLD 4 (*n* = 28) | 382.9 (528.2–687.4) | 400.3 (575.8–778.6) | 26.6 (29.2 – 167.0) | 36.7 (73.6–96.4) |
| | *p* ** | 0.734 | 0.228 | 0.588 | 0.359 |
| Treatment outcome | Good (*n* = 67) | 379.2 (482.9–690.9) | 290.4 (485.4–645.7) | 25.6 (53.5–178.7) | 29.1 (51.4–91.3) |
| | Death (*n* = 30) | 360.5 (434.7–669.0) | 328.1 (573.1–788.8) | 24.6 (26.2–31.7) | 33.7 (64.1–106.7) |
| | *p* * | 0.389 | 0.751 | 0.613 | 0.287 |

* Mann–Whitney test, ** Kruskal–Wallis.

## 4. Discussion

In our study, the patients were mainly elderly with severe exacerbations and complications. This feature might be related to the selection criteria of study patients, which were patients with COPD admitted to central hospitals, with a critical illness [21]. We also recorded that the figures for sputum bacterial culture in patients were low (27.8%). Previous studies have shown that the result of sputum culture depends mainly on antibiotic use before admission [9]. Moreover, because the majority of patients in our study used antibiotics before sputum culture, the results of sputum culture were low. The rate of patients dying during exacerbations was high (30.9%). This may be caused by the studied patients being mainly elderly with severe exacerbations and many complications.

In our study, mean levels of CRP and PCT both increased, while total IgG, IgG1, and IgG3 decreased in patients with COPD. There was a negative correlation between significantly elevated PCT levels and IgG3 at the time of admission. This result reflects a patient's inflammatory response increasing during the exacerbation. During COPD exacerbations, especially infectious exacerbations, there **is** a local amplification of inflammatory response

(bronchi and lung parenchyma) combined with systemic inflammation, which increases inflammatory markers such as CRP and PCT [22–24]. Hypogammaglobulinemia was a risk of exacerbation and mortality in patients with COPD [25]. Furthermore, the addition of intravenous Ig therapy to the treatment of infectious exacerbations in group D patients with COPD should be considered [26].

IgG is a class of antibodies that accounts for 75–80% of the total antibodies in the serum and plays an important role in respiratory infections [25,27]. There are four IgG subclasses with different roles in the body's defense mechanism against infection. Of those, IgG1 accounts for the highest proportion and has a major role in protecting the body against protein-derived antigens. IgG2 plays a role in resistance to polysaccharide-coated microorganisms. IgG3 plays a major role in the immune response against respiratory viruses, and IgG4 takes a leading role in helping the body fight against respiratory infections [28,29]. Our study showed that the mean levels of total IgG in patients at times T1 and T2 were significantly lower than those of the control group. Regarding IgG subclasses, IgG1, IgG3, and IgG4 levels at times T1 and T2 were significantly lower than those in the control group, and the IgG3 levels at time T1 were significantly higher than those at time T2. Thus, there was a decrease in serum total IgG and IgG1, IgG3, and IgG4 levels in patients during exacerbations compared with healthy subjects. Some previous studies about serum IgG levels in patients with stable COPD found that levels of total IgG and some IgG subclasses decreased [15,17,30,31]. A study on 59 adult patients with asthma and stable COPD in Korea found that IgG subclass deficiency was the most common phenotype (67%), followed by total IgG deficiency (20%), in which IgG3 and IgG4 were the most affected subclasses [15]. Low serum IgG levels were associated with an increased risk for recurrent acute exacerbations, lower respiratory infections, a decline in lung function, and hospitalization [25,32]. However, the IgG3 levels at time T1 were significantly higher than those at time T2. In healthy individuals, serum IgG levels are elevated during respiratory infections with a mean biological half-life of 3 weeks, so they change slowly as the infection resolves [28]. The inflammatory process in COPD goes through three phases: initiation, progression, and consolidation [33]. During the exacerbation of the disease, especially the infection-derived exacerbation, there **is** an increase in the inflammatory process. Inflammation-induced factors include both pathogen-associated molecular pattern molecules (PAMPs) and damage-associated molecular patterns (DAMPs) [28,33]. Our study showed that in patients with COPD, during the initiation of infectious exacerbation, there was also a remarkable immunodeficiency (reduction in total IgG, IgG1, and IgG3). According to our knowledge, there has not been a study that points out a decrease in IgG levels in COPD exacerbations; however, there may be an association between reduced IgG levels in stable COPD and exacerbations.

The mean levels of total IgG and IgG subclasses when the patients were at time T1 did not differ according to the treatment outcomes and results of sputum bacterial culture ($p > 0.05$). Leitao Filho et al. (2018) evaluated serum Ig changes in patients with stable COPD and found that approximately one in five patients with COPD had one or more IgG subclass deficiencies. Reduced IgG subclass levels were an independent risk factor for both COPD exacerbations (IgG1 and IgG2) and hospitalizations (IgG2) in two COPD cohorts [29]. IgG2 deficiency presents with impaired polysaccharide responses leading to a higher susceptibility to infections caused by encapsulated pathogens (*Streptococcus pneumoniae* and *Haemophilus influenza type B*), which was frequently implicated in bacterial COPD exacerbations [34,35]. Immune deficiency, especially IgG deficiency, is associated with recurrent infections in patients with chronic airway diseases such as asthma, COPD, or bronchiectasis. IgG3 accounts for approximately 2–4% of total serum IgG, is the most common deficiency subclass in chronic airway diseases, and is strongly associated with the risk of recurrent infections, as well as exacerbations. It is possible to have an IgG subclass deficiency without a general IgG deficiency, or a combined deficiency of multiple subclasses [15,36].

The median IgG3 levels were significantly higher in patients with non-life-threatening exacerbation compared with those in patients without it ($p < 0.05$). IgG1 levels were significantly higher in the group with respiratory failure than in the counterpart ($p = 0.01$). IgG3 plays an important role in controlling and protecting against respiratory infections, especially bacterial infections [37,38]. The bacteriostatic activity of IgG3 is greater than that of IgG1 [39]. Holm et al. (2020) found that hypogammaglobulinemia might be involved in poor outcomes in COPD [25]. In a study by Kim et al. (2016) in Korea with 59 adult patients with bronchial asthma and COPD, IgG3 deficiency was the most common (88.1%) [15]. The results of our study indicated that the serum IgG3 level in the patients with non-life-threatening exacerbation was significantly higher than that in the life-threatening group; thus, IgG3 has a major role in the infectious exacerbation of COPD.

Our study has limitations. This was a prospective study with a small sample size. Treatment of patients with corticosteroids may affect the results of the analysis of Ig levels after treatment. Moreover, we did not evaluate other biomedical markers which affected Ig concentrations. In the near future, we plan to study and evaluate some more interleukins in combination with Ig to help prognosis for patients with severe exacerbation of COPD in clinical practice.

## 5. Conclusions

In conclusion, in group D patients with COPD with infectious exacerbations, there was a decrease in the total serum IgG, IgG1, and IgG3 levels. IgG1 and IgG3 levels were associated with complications of respiratory failure and the severity of COPD exacerbation, which was important in disease prognosis.

**Author Contributions:** Conceptualization, T.B.T. and B.N.D.; methodology, T.B.T. and B.N.D.; software, S.T.N. and T.D.L.; validation, T.B.T. and B.N.D.; formal analysis, T.B.T., B.N.D., T.D.L. and S.T.N.; investigation, T.B.T. and B.N.D.; resources, T.B.T., T.T.V., K.X.N. and B.N.D.; data curation, T.B.T., B.N.D., C.H.N., H.N.V. and S.T.N.; writing—original draft preparation, T.B.T. and B.N.D.; writing—review and editing, T.B.T., T.T.V., K.X.N., C.H.N., H.N.V., T.D.L., S.T.N., H.K.D., N.K.T.P. and B.N.D.; visualization, N.K.T.P.; supervision, B.N.D.; project administration, T.B.T. All authors made substantial contributions to the conception and design, acquisition of data, or analysis and interpretation of data; took part in drafting the article or revising it critically for important intellectual content; gave final approval of the version to be published; and agree to be accountable for all aspects of the work. All authors have read and agreed to the published version of the manuscript.

**Funding:** The authors received no financial support for the research, authorship, and/or publication of this article.

**Institutional Review Board Statement:** The protocol was approved by Ethics Committee of Military Hospital 103 (No. 57/2014/VMMU-IRB). The study was also conducted using good clinical practice following the Declaration of Helsinki.

**Informed Consent Statement:** All participants provided written informed consent. Informed consent was obtained from all subjects involved in the study.

**Data Availability Statement:** The data supporting this research are available from the authors on reasonable request.

**Conflicts of Interest:** The authors declare no conflict of interest.

## Abbreviations

COPD, chronic obstructive pulmonary disease; CRP, C-reactive protein; DAMPs, damage-associated molecular patterns; GOLD, Global Initiative for Chronic Obstructive Lung Disease; Ig, immunoglobulin; PAMPs, pathogen-associated molecular patterns; PCT, procalcitonin; TNF-$\alpha$, tumor necrosis factor-$\alpha$; VMMU, Vietnam Military Medical University.

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
