# Peer review of "Changes in Serum Immunoglobulin G Subclasses during the Treatment of Patients with Chronic Obstructive Pulmonary Disease with Infectious Exacerbations"

_arm, doi:10.3390/arm90060056_

Round 1

Reviewer 1 Report

This article aimed exploring the changes in serum Ig levels and their relationship with outcomes of acute infectious exacerbations in COPD patients and found IgG3 levels were associated with the severity of COPD exacerbation. However, further clarification and justification regarding the methodology, analyses and reporting, interpretation/discussion of results is required.

1. Please provide the characteristics of healthy participants. The comparison between healthy and exacerbations in COPD participants should be added in the current analysis.

2. The specific description of characteristics of patients should be showed in the results part, not the discussion part.

3. Some other potential factors should be mentioned. Such as smoking status and pulmonary function.

4. The discussion part should emphasize the main result of current analysis and discuss potential possible mechanisms, point the future research direction and clinical significance.

5. Please pay attention to the accuracy of words spelling and punctuations using.

 6. There is much room for improvement of the English writing and article format.

Author Response

Dear Reviewer 1,

Many thanks for your comment on our manuscript “Changes in serum immunoglobulin G subclasses during the treatment of patients with chronic obstructive pulmonary disease with infectious exacerbations”.

Our team has reviewed all the suggestions critically and revised the paper accordingly. We are grateful for the detailed comments we received, which we used to improve the paper.

Please find attached file our responses to your comment.

Reviewer 2 Report

I have read the article by Ta et al. with great interest. The authors investigated blood IgG levels in patients with COPD.

Comments:

·       Instead of COPD patients please use patients with COPD.

·       Abstract. Please, add patient and control numbers and who comprised the control group.

·       Treatment outcomes. How many patients were readmitted following discharge? Usually, this is classified as a treatment failure. Please, comment.

·       Statistical analysis. Please, provide power calculations.

·       Results. Please, provide comparisons between the COPD and control groups in terms of age, BMI, smoking history, comorbidities (which were not excluded, such as hypertension).

·       Results. Table 1. Please, provide blood gas results.

·       Results. Table 1. 27.8% had positive sputum culture. Please, let us know these organisms.

·       Results. Do you have data on full blood count (i.e. WBC, neutrophil%, eosinophil%)?

·       Results. Please, add data on long term treatment.

·       Results. Please, add data on the treatment for exacerbation (i.e. drugs, O2, NIV, IMV).  

Author Response

Dear Reviewer 2,

Many thanks for your comment on our manuscript “Changes in serum immunoglobulin G subclasses during the treatment of patients with chronic obstructive pulmonary disease with infectious exacerbations”.

Our team has reviewed all the suggestions critically and revised the paper accordingly. We are grateful for the detailed comments we received, which we used to improve the paper.

Please find attached file our responses to your comment.

Reviewer 3 Report

Dear author's

I have reviewed your manuscript and I have the following comments: 

Please explain if IgG is specific marker for COPD?

There are some clinical/or associate diseases that increase the IgG level? This scenario can create bias.

The sample of the study is very small. Prospective randomised studies are necessary in order to draw conclusion.

In the discussion section is mandatory to compare your results with the existing literature.

Please explain the novelty of your study.

Minor English edits.

Author Response

Dear Reviewer 3,

Many thanks for your comment on our manuscript “Changes in serum immunoglobulin G subclasses during the treatment of patients with chronic obstructive pulmonary disease with infectious exacerbations”.

Our team has reviewed all the suggestions critically and revised the paper accordingly. We are grateful for the detailed comments we received, which we used to improve the paper.

Please find attached file our responses to your comment.

Round 2

Reviewer 1 Report

I agree to accept this manuscript after minor review by editor.

Author Response

Dear Reviewer 1,

Many thanks for your comment on our manuscript “Changes in serum immunoglobulin G subclasses during the treatment of patients with chronic obstructive pulmonary disease with infectious exacerbations”.

Our team has reviewed all the suggestions critically and revised the paper accordingly. We are grateful for the detailed comments we received, which we used to improve the paper.

Please find below our responses to your comment.

Reviewer 2 Report

The article is still below the standard; however there was some improvement. The authors have not addressed most of my comments. Most particularly:

1. The authors did not provide detailed comparisons between the patient and control groups (Table 1).

2. Was there any correlation between Ig levels and clinical characteristics (i.e. lung function, blood gases, sputum microbiology)?

3. How many patients were treated with antibiotics and/or steroids?

Author Response

Dear Reviewer 2,

Many thanks for your comment on our manuscript “Changes in serum immunoglobulin G subclasses during the treatment of patients with chronic obstructive pulmonary disease with infectious exacerbations”.

Our team has reviewed all the suggestions critically and revised the paper accordingly. We are grateful for the detailed comments we received, which we used to improve the paper.

Please find below our responses to your comment.

Reviewer 3 Report

Dear author’s

Thank you for your respinse.

If your intention is to continue the study why do you want to publish partial results?

Author Response

Dear Reviewer 3,

Many thanks for your comment on our manuscript “Changes in serum immunoglobulin G subclasses during the treatment of patients with chronic obstructive pulmonary dis-ease with infectious exacerbations”.

Our team has reviewed all the suggestions critically and revised the paper accordingly. We are grateful for the detailed comments we received, which we used to improve the paper.

Please find below our responses to your comment.

Round 3

Reviewer 2 Report

The authors did not address important points. Some patients did others did not recieve systemic cortocosteroids. As this therapy influences Ig levels, we need to have sugbroup analyses based on corticosteroid use, dose and duration. This information is repeatedly not told by the authors.

Author Response

(The authors gave the same response as above.)

Reviewer 3 Report

Thank you for your response.

Author Response

Dear Reviewer 3,

Many thanks for your comment on our manuscript “Changes in serum immunoglobulin G subclasses during the treatment of patients with chronic obstructive pulmonary disease with infectious exacerbations”.

Our team has reviewed all the suggestions critically and revised the paper accordingly. We are grateful for the detailed comments we received, which we used to improve the paper.

Please find below our responses to your comment.
